# Urinary Continence Resolution after Bariatric Surgery: Long-Term Results after Six-Year Follow-Up

**DOI:** 10.3390/jcm12062109

**Published:** 2023-03-08

**Authors:** Thibaut Waeckel, Khelifa Ait Said, Benjamin Menahem, Anais Briant, Arnaud Doerfler, Arnaud Alves, Xavier Tillou

**Affiliations:** 1Urology and Transplantation Department, University Hospital of Caen, 14000 Caen, France; 2Department of Digestive Surgery, University Hospital of Caen, 14000 Caen, France; 3Registre des Tumeurs Digestives, INSERM UMR 1086 ANTICIPE, 14032 Caen, France; 4Department of Biostatistics, University Hospital of Caen, 14000 Caen, France; 5Pôle de Formation et de Recherche en Santé, 2 rue des Rochambelles, 14033 Caen, France

**Keywords:** urinary incontinence, bariatric surgery

## Abstract

Background: Bariatric surgery is known to improve stress urinary incontinence (SUI) and overactive bladder disorders (OAB). However, there is little long-term follow-up in the literature. Objective: To determine the long-term effect of bariatric procedures on SUI and OAB and their impact on quality of life, we applied the ICIQ and USP questionnaires. Setting: The research was conducted at a French university hospital with expertise in bariatric surgery. Methods: We performed an updated follow-up at 6 years of a prospective cohort of 83 women who underwent a bariatric procedure between September 2013 and September 2014. The women completed the USP and ICIQ questionnaires before surgery, 1 year and 6 years after the surgery. Results: Of the 83 patients, 67 responded (80.7%) in full. SUI remained improved at 6 years: the USP score decreased from 3 [1; 7] before surgery to 0 [0; 1] (*p* = 0.0010) at 1 year after surgery and remained at 0 [0; 0] (*p* = 0.0253) at 6 years. The decrease in the OAB symptom score remained statistically significant: 3 [1; 7] before the surgery vs. 2 [0; 5] at 6 years (*p* = 0.0150). However, this improvement was significantly less pronounced than at 1 year: 0 [0; 1] (*p* = 0.002). Conclusions: Bariatric surgery seems to be effective at treating SUI and OAB with a long-lasting effects, still noted at 6 years.

## 1. Introduction

Obesity is a worldwide public health problem affecting 13% of the population [1]. The Obepi 2 study [2] published in 2020 shows that overweight affects almost half of the population in France. Bariatric surgery can be considered for patients with a body mass index (BMI) ≥ 40 kg/m^2^, or ≥35 kg/m^2^ when associated with another comorbidity condition (after of 6 to 12 months of medical treatment) [3].

Urinary incontinence is defined according to the terminology of the International Continence Society as “involuntary loss of urine through the urethra”, constituting a social or hygiene problem that can be objectively demonstrated [4]. There are three main types of urinary incontinence: stress urinary incontinence (SUI); overactive bladder (OAB) and mixed incontinence. It is a major public health problem; in France, approximately three million women are affected by urinary incontinence [5]. After failure of medical treatment, sub-urethral slings can be proposed to treat SUI in women. For women with a BMI ≥ 35 kg/m^2^, the results of sub-urethral slings are poor: only 81% achieve continence vs. 96% in the general population [6]. Weight loss for obese women who suffer from urinary incontinence is recommended too by the European Association of Urology (EAU) [7]. 

We established previously, with a one-year follow-up, that weight loss after bariatric surgery improved stress urinary incontinence (SUI), OAB (overactive bladder), dysuria and quality of life (QoL). These results were especially convincing in women [8].

The aim of this study was to determine the impact of bariatric surgery on female urinary incontinence at 6 years.

## 2. Methods

### 2.1. Endpoints

The primary endpoint was the evolution of urinary incontinence at 6 years, as evaluated by the USP questionnaire. The secondary endpoints were the evolution of urinary incontinence and associated quality of life between 1 and 6 years.

### 2.2. Patients

Between September 2013 and September 2014, in a French University hospital with expertise in bariatric surgery, 83 women underwent bariatric procedures. The surgical technique chosen (Roux-en-Y gastric bypass (LRYGB) or sleeve gastrectomy (LSG)) was discussed for each patient during a multidisciplinary meeting. All patients were included in an observational prospective cohort [8]. They completed two urinary questionnaires: the Urinary Symptom Profile (USP) and the International Consultation on Incontinence Questionnaire (ICIQ).

### 2.3. Questionnaires

The USP (Appendix A) was developed by the French Association of Urology in 2005. The 13 questions explore all urinary symptoms: stress urinary incontinence (SUI), overactive bladder (OAB) and obstructive symptoms. The ICIQ (Appendix A) focuses on SUI, urge incontinence, mixed incontinence and their impact on quality of life (QoL) [9]. The ICIQ is validated in French. These questionnaires are complementary: the USP does not assess QoL, while the ICIQ does not assess OAB symptoms. The higher the score, the more severe the symptoms. The endpoint of this study was based on these questionnaires. The questionnaires were completed in the hospital preoperatively, by mail at one year and by phone at six years.

### 2.4. Weight Loss Evaluation

Weight loss was expressed as the change in body mass index (BMI), the percentage of excess weight loss (%EWL) and the percentage of total weight lost (%TWL). The %EWL is based on an ideal body weight equivalent to a BMI of 25kg/m^2^. The %TWL is based on the maximum weight reached by the patient.

### 2.5. Statistical Analysis

Baseline characteristics were described by numbers (percentages) for qualitative variables and the median (and interquartile range (IQR)) for quantitative variables. The normal distribution of quantitative variables was tested using the Shapiro–Wilk test. The primary endpoint and secondary endpoints were analyzed using Friedman’s non-parametric test to compare repeated measures. Wilcoxon’s rank test was used to compare the %EWL at one and six years. Spearman’s correlations were generated to assess the association between the evolution of SUI and evolution of %EWL and %TWL. Post hoc analysis was performed with Dunn’s multiple comparisons.

Some secondary endpoints (BMI, ICIQ, %EWL, %TWL) had missing data; at most, there were 3 missing data (4%) for each of the 67 patients. We handled missing data using multiple imputations through the fully-conditional specification method, based on the age and values of the secondary endpoints at baseline, with the generation of one dataset.

Subgroup analyses were performed for each surgical modality (LSG and LRYGB). We compared the evolution before surgery vs. at six years of IMC, % TWL and USP and ICIQ scores using the Mann–Whitney U test. 

GraphPad Prism version 9.3.1 (GraphPad Software, San Diego, CA, USA, https://www.graphpad.com accessed on 3 January 2023) and SAS software version 9.4 (SAS institute, NC, Cary, USA) were used for statistical analyses and a *p*-value < 0.05 was considered to denote statistical significance.

### 2.6. Ethics 

The study was conducted according to the guidelines of the Declaration of Helsinki. Data collection followed French legislation concerning prospective non-interventional studies to evaluate routine care (Article Art.L1121-1-2 of French Public Health Code). Data were collected from a prospectively maintained database of patients who underwent laparoscopic bariatric surgery at the University Hospital of Caen, a French specialized and accredited bariatric center, after January 2012 (CNIL 2204611v0). The study did not require submission to a consultative committee for persons’ protection in biomedical research.

## 3. Results 

### 3.1. Characteristics of the Population 

The median age of the women operated on was 46.0 [36.0; 54.0] years. The preoperative BMI was 42.5 [39.4; 46.0] kg/m^2^. These women were suffering from sleep apnea (29–43.3%), gastric reflux (7–11.4%), arthrosis (13–19.4%), diabetes (14–20.9%), arterial hypertension (22–32.8%) and dyslipidemia (10–13.9%) (Table 1).

### 3.2. Follow-Up Questionnaires 

Both questionnaires were completed by 83 women (19 LSG–64 LRYGB) at one year and by 67 (80.7%) women at six years (14 LSG–53 LRYGB). Two women died (2.4%) and 14 were lost to follow-up (16.9%). After 6 years of follow-up, 22 patients still attended regular in-office follow-up (26.5%). The mean age was 45.7 ± 11.6 years old. A flow chart of the follow-up is shown in Figure 1. 

### 3.3. Weight Loss

Weight loss at 6 years remained significant despite weight regain between the first and sixth years: %EWL increased from 73.0 [59.0; 94.0] to 61.0 [45.0; 80.0] (*p* < 0.0001); %TWL increased from 36.1 [31.6; 44.4] to 29.9 [23.9; 39.2] (*p* < 0.0001). BMI changed from 42.5 [39.4; 46.0] kg/m^2^ before the surgery to 29.1 [26.0; 33.3] kg/m^2^ one year after the surgery, then to 30.8 [28.1; 36.4] kg/m^2^ six years after the surgery (*p* < 0.0001).

No significant difference in evolution was found between pre-surgery and six years post-surgery according to the type of surgery (LSG or LRYGB) for BMI (*p* = 0.57) or %TWL (*p* = 0.51)

### 3.4. Stress Urinary Incontinence

SUI remained improved at 6 years: the USP score decreased from 1 [0; 3] before surgery to 0 [0; 0] (*p* = 0.001) at 1 year after surgery and 0 [0; 0] (*p* = 0.0253) at 6 years. There was no deterioration of the USP score between one and six years (*p* > 0.9999). There was no correlation between the evolution of SUI and the evolution of %EWL (r = −0.20; 95 CI [−0.42; 0.05]; *p* = 0.10) or %TWL (r= −0.10; 95% CI [−0.34; 0.15]; *p* = 0.40). The type of surgery (LSG or LRYGB) did not influence the evolution between the results before surgery and at six years (*p* = 0.72).

### 3.5. Overactive Bladder

The decrease in the OAB symptom score remained statistically significant: 3 [1; 7] before the surgery vs. 2 [0; 5] at six years (*p* = 0.0150). However, this improvement was significantly less pronounced than at one year: 0 [0; 1] (*p* = 0.002). There was no correlation between the evolution of OAB and the evolution of %EWL (r = −0.05; 95% CI [−0.30; 0.20]; *p* = 0.67) or %TWL (r = −0.05; 95% CI [−0.29; 0.20]; *p* = 0.69). The variation of the score before surgery vs. at six years was not influenced by the type of surgery (*p* = 0.12).

### 3.6. Dysuria

Despite weight loss, dysuria was not improved at one and six years. The score was 0 [0; 0] before the surgery and was still 0 [0; 0] at one year (*p* = 0.8997) and 0 [0; 0] at six years (*p* = 0.7306).

### 3.7. Quality of Life Related to Urinary Symptoms

The improvement in QoL related to urinary symptoms at one year (0 [0; 0] vs. 3 [0; 9] (*p* = 0.0001)) was not found at six years: 0 [0; 7] (*p* = 0.3918). There was a significant deterioration of the ICIQ (*p =* 0.0325). The results before surgery vs. at six years did not differ significantly depending on the type of surgery (*p* = 0.39).

The questionnaire results and evolution of dimensions are show in Table 2.

## 4. Discussion

To our knowledge, with 66 months of follow-up, this study offers data with the longest follow-up in the literature. Recent meta-analyses on this topic reported follow-ups of approximately 1–2 years [10,11,12]. Follow-up of bariatric surgery patients is difficult due to moderate compliance with care. In the SOS study, the rate of follow up at 10 years was 42% [13]. The follow-up of patients in this study appeared to be consistent with those reported in the literature. 

Weight loss after bariatric surgery was shown to improve SUI [14]. Our study confirmed this effect in the long term despite a partial weight gain. SUI in obese patients is due to abdominal pressure exceeding the strength of the urinary sphincter. Reducing intra-abdominal weight may be sufficient to improve SUI. 

Despite an improvement in the OAB at one year, our study demonstrated a deterioration at six years. This degradation did not seem to be related to %EWL or %TWL. The production of leptin by the adipose tissue leads to stimulation of the noradrenergic sympathetic nerves [15]. Perivesical fat accumulation is also associated with local inflammation leading to OAB [16]. Furthermore, researchers assert that the distribution of fat in women could explain this phenomenon, although it is recognized that weight gain is more often localized in the hips than in the perivesical area.

The EPICONT [17] study, a large Norwegian prospective study, showed a dose effect of weight in urinary incontinence without distinguishing the type of surgery. This effect seems to be cumulative because we did not find any correlation with %TWL. Rapid weight loss could provide a better improvement in urinary incontinence [18]. This hypothesis tends to give an advantage to LRYGB; however, in this cohort, the type of surgery (LSG or LRYGB) did not seem to influence the outcome at six years. This can probably be explained by a lack of power. Indeed, only 14 women were treated with LSG. 

The strengths of our study were its prospective nature and the number of patients included, with a long follow-up associated with a good response rate of completed questionnaires at one year. The questionnaire at one year was administered by email and had a self-administrated design, which may have supported the good response rate. In addition, weight was measured at a one-year follow-up clinical consultation. 

This study also had some limitations. To simplify and optimize data collection and minimize the number of people lost to follow-up, the collection of weight measurements and the answers to the questionnaires at six years were carried out by phone. The declared weight was, therefore, subject to social desirability bias [19]. This bias could have minimized the reported weight gain at six years. Furthermore, USP and ICIQ are validated, self-administered questionnaires [20,21], but phone responses to them may have biased the results. However, considering the number of patients lost to long-term follow-up, this method allowed us to increase the amount of data. Additional points to note are that since 2012, the Caen University Hospital has been recognized as a Specialized Obesity Center (CSO); there was thus a risk of selection bias given the center’s expertise. Finally, this was a retrospective study on a prospective cohort; there were, therefore, biases inherent in the study’s design.

## 5. Conclusions

Obesity is a fast-growing public health issue around the world, and related urinary and fecal disorders are underestimated. These results support a long-term effect of bariatric surgery (LSG or LRYGB) on urinary incontinence in women. This effect seems to diminish in the long term compared to at one year but is still notable. The improvement does not seem to correlate with %EWL or %TWL. Management of SUI and OAB in obese patients should require them to join a bariatric surgery program. 

## Figures and Tables

**Figure 1 jcm-12-02109-f001:**
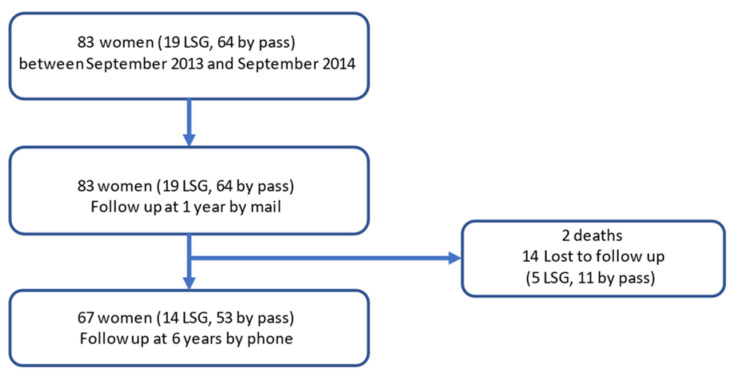
Flow chart of follow-up.

**Table 1 jcm-12-02109-t001:** Study population (number of patients (*N*), mean +/− standard deviation, minimum, first interquartile, median, third interquartile, maximum).

Variable	Population (*n* = 67)
Age	46.0 [36.0; 54.0]
BMI (kg/m^2^)	42.5 [39.4; 46.0]
Sleep apnea	29 (43.3%)
Gastric reflux	7 (11.4%)
Arthrosis	13 (19.4%)
Diabetes	14 (20.9%)
HTA	22 (32.8%)
Dyslipidemia	10 (13.9%)

**Table 2 jcm-12-02109-t002:** Evolution of weight and score results for each questionnaire and dimension (median (IQR)).

Questionnaires and Dimensions							
Variable	Before Surgery (A)	A vs. B°	1 Year after Surgery (B)	B vs. C°	6 Years after Surgery (C)	A vs. C°	*p*-Value *
BMI (kg/m^2^)	42.5 [39.4; 46.0]	*p* < 0.0001	29.1 [26.0; 33.3]	*p* = 0.0009	30.8 [28.1; 36.4]	*p* < 0.0001	*p* < 0.0001
%EWL			73.0 [59.0; 94.0]	*p* < 0.0001 ^#^	61.0 [45.0; 80.0]		
%TWL	6.7 [3.1; 11.5]	*p* < 0.0001	36.1 [31.6; 44.4]	*p* < 0.0001	29.9 [23.9; 39.2]	*p* < 0.0001	*p* < 0.0001
USP stress urinary incontinence	1 [0; 3]	*p* = 0.0010	0 [0; 0]	*p* > 0.9999	0 [0; 0]	*p* = 0.0253	*p* < 0.0001
USP overactive bladder	3 [1; 7]	*p* = 0.0001	0 [0; 1]	*p* = 0.0002	2 [0; 5]	*p* = 0.0150	*p* < 0.0001
USP dysuria	0 [0; 0]	*p* = 0.8997	0 [0; 0]	*p* > 0.9999	0 [0; 0]	*p* = 0.7306	*p* = 0.0077
ICIQ	3 [0; 9]	*p* = 0.0001	0 [0; 0]	*p* = 0.0325	0 [0; 7]	*p* = 0.3918	*p* < 0.0001

Post hoc analyses with Dunn’s multiple comparison. ^#^ Wilcoxon’s rank test. * Friedman’s test.

## Data Availability

Not applicable.

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
