# Peer review of "Urinary Continence Resolution after Bariatric Surgery: Long-Term Results after Six-Year Follow-Up"

_jcm, 2023, doi:10.3390/jcm12062109_

Round 1

Reviewer 1 Report

Dear Authors, 

This is a very interesting paper related to bariatric surgery is known to improve stress urinary incontinence (SUI) and overactive bladder disorders (OAB).  This is an interesting topic that is not always well covered in the literature. In fact, not so many references are related to this type of disease, related to obesity. Furthermore, we have to understand that bariatric surgery plays a role in all this improvements.

The authors addressed the topic in a correct way by means of a retrospective study. The authors show exactly the objective of the study and the study is well conducted.

With my English spoken language, I think the paper is well written, however. It has to be confirmed with a native English speaker. To my reading, the paper is clear and easy to read. It is well constructed and also has very defined parts that make the lecture very simple.

According to the methodology, the conclusions are consistent with the evidence and arguments presented. Although the number of cases could be considered limited, the evidence with these patients is correct and shows the conclusions.

This paper is interesting compared to other available literature because it shows a long follow-up of more than 66 months. Results are consistent with the appears and reported in the literature.

Coments: 

%EWL is a percentage not rated in Kg.  (Table)

Thank you. 

Author Response

Dear reviewer, here are our responses (cf upload file).

Yours sincerely,

Benjamin MENAHEM MD, PhD

Reviewer 2 Report

More specific comments are as follows:

1. Study aim has to be formulated more precisely

2. The subsection on statistical analysis has to be expanded:

i. Did the authors test for the normality of data distribution? If yes, how they did this?

ii. there are no such things as "first interquartile" and "third interquartile"

iii. In which cases the Friedman non-parametric test was applied? Please, justify this choice.

iv. Why the ANOVA was selected for comparison of the surgery results at different time points? ANOVA is best selected for  normally distributed independent variables, which was not the case.

So, the authors need to reconsider all statistical tests used and re-run the statistical analysis. There is such a mess. The updated statistical analysis might impact the conclusions drawn.

3. You need to consider the results of each surgical modality (LSG and LRYGB) separately. Was one intervention superior to another?

4. The authors stated correlation analysis but these findings are not presented.

5. I never ever saw such a short discussion for a study with so important findings. The authors need to consider their findings in all detail.

6. The conclusion section is messy and is not supported by the findings. Avoid making far-going conclusions, focus at interpretation of your findings.

Author Response

Dear reviewer

Here are our responses to your comments. (upload file)

Yours sincerely,

Benjamin MENAHEm MD PhD

Round 2

Reviewer 2 Report

Well done!